# High Energy Density of Ball-Milled Fluorinated Carbon Nanofibers as Cathode in Primary Lithium Batteries

**DOI:** 10.3390/nano14050404

**Published:** 2024-02-22

**Authors:** Marie Colin, Elodie Petit, Katia Guérin, Marc Dubois

**Affiliations:** Clermont Auvergne INP, CNRS, Institut de Chimie de Clermont-Ferrand (ICCF UMR 6296), BP 10448, Université Clermont Auvergne, F-63000 Clermont-Ferrand, France; mariecolin03@gmail.com (M.C.); elodie.petit@uca.fr (E.P.); katia.araujo_da_silva@uca.fr (K.G.)

**Keywords:** primary lithium battery, sub-fluorination, ball-milling, carbon nanofibers, power density

## Abstract

Sub-fluorinated carbon nanofibers (F-CNFs) can be described as a non-fluorinated core surrounded by a fluorocarbon lattice. The core ensures the electron flux in the cathode during the electrochemical discharge in the primary lithium battery, which allows a high-power density to be reached. The ball-milling in an inert gas (Ar) of these F-CNFs adds a second level of conductive sp^2^ carbons, i.e., a dual sub-fluorination. The opening of the structure changes, from one initially similar multi-walled carbon nanotube to small lamellar nanoparticles after milling. The power densities are improved by the dual sub-fluorination, with values of 9693 W/kg (3192 W/kg for the starting material). Moreover, the over-potential of low depth of discharge, which is typical of covalent CFx, is suppressed thanks to the ball-milling. The energy density is still high during the ball-milling, i.e., 2011 and 2006 Wh/kg for raw and milled F-CNF, respectively.

## 1. Introduction

Because fluorinated carbons (CFxs) exhibit the highest theoretical energy density among current cathode materials for lithium primary batteries, extensive efforts are devoted to the enhancement of either their energy density, their power density, or both (see recent review papers [1,2,3,4]). The very recent strategies involve (i) considering new carbonaceous precursors for fluorination or innovative fluorination routes [5,6,7,8,9,10,11,12,13,14], (ii) chemical changes in the CFx, either on the surface or in the bulk [15,16,17,18,19,20,21,22], and (iii) change in the electrolyte composition [13,23,24,25,26,27,28,29]. Ball milling, a simple process for reducing the particle size of a compound, is widely used for electrode materials [30,31,32,33,34], creating more space to accumulate the LiF(s) particles formed. Some milling treatments of fluorinated carbons under different conditions, with or without additives, have led to improved electrochemical performance [35,36]. The milling of CFx in the presence of urea has made it possible to reduce particle size and increase inter-layer spacing. The defluorination caused by milling leads to better electronic conduction, which reduces the potential drop at the start of the electrochemical process without loss of capacity. The increase in discharge potential leads to improved electrochemical performance, reaching a maximum power density of 10,309 W/kg [34]. Reddy et al. have shown that simple mechanical ball milling of fluorinated graphite improves power densities, with 10,000 W/kg delivered at a current rate of 6C, whereas before milling fluorinated graphite delivered no capacity at this rate [35]. The use of nanometric fluorinated carbons has already produced good results in lithium batteries. Ahmad et al. achieved a very high capacity of 1180 mAh/g by using fluorinated carbon nanodisks with both a low CF_2_ and CF_3_ group content and a ‘reinforcement’ effect by the core of the matrix (a few disks are maintained at the core of the matrix after electrochemical reduction); this makes it possible to achieve an ‘extracapacity’, i.e., a capacity greater than the theoretical one, involving a new electrochemical mechanism in addition to the one conventionally described [36]. Multi-walled fluorinated carbon nanotubes (F/C = 0.81) showed improved energy and power densities with 2007 Wh/kg and 3861 W/kg delivered in a lithium battery [37]. Fluorinated carbon nanocapsules (x > 1 in CFx composition) with a hollow 3D structure and a CF_3_-poor surface delivered a capacity of 1056 mAh/g and an energy density of up to 2487 Wh/kg [38].

When a cell with CFx as cathode material is discharged, the Li^+^ ions combine with the F^−^ ions to form the LiF(s) compound, which is incorporated into the carbon matrix at the edges [39,40]. This causes partial exfoliation and volume expansion of the carbon matrix [41]. These phenomena damage the physical integrity of the electrode, causing part of the active material to lose electrical contact even before it has been fully discharged, and the electrochemical performance is thus degraded. The accumulation of LiF at the edges of the sheets also limits and slows down access to all the active C-F bonds available for electrochemical reduction, once again reducing electrochemical performance. High grain size and structural order will amplify this accumulation of LiF at the planar edges. To counter this problem, the active material can be modified while directing the growth of the LiF(s) formed. The first line of research involves working on exfoliated fluorinated graphite, which will already have an open structure to accommodate LiF growth. In addition, the lower packing density of the sheets naturally induces smaller LiF crystals at the edges of the plane. Uncontrolled volume expansion due to the discharge could then be avoided by maintaining the integrity of the electrode. Previous works by Mar et al. on the fluorination of expanded graphite were aimed at exploring this strategy by obtaining a material with cavities enabling LiF [42].

In the present work, the aim is to combine sub-fluorination and milling to exploit the effects of both on electronic conduction, thanks to the non-fluorinated core and decrease in grain size. Sub-fluorinated carbon nanofibers (CNF) with a composition of CF_0.8_ have been selected as the starting material for the ball-milling. Moreover, micrometric sub-fluorinated graphite with the same composition is considered for comparison purposes to evidence the effect of the nanometric size of the CNF.

## 2. Materials and Methods

### 2.1. Materials

High purity (>90%) carbon nanofibres with 80–200 nm diameters and 2–20 μm lengths have been furnished by the MER Corporation. They were obtained using the CVD method and then post-treated under an argon atmosphere at 1800 °C to increase their graphitization degree. KS4 graphite with a 4 μm grain size was provided by Timcal Co., Ltd., Bodiio, Switzerland. The fluorinations of CNF and KS4 graphite were performed in pure F_2_ gas for temperatures of 450 and 550 °C, respectively. Details on the synthesis and direct fluorination mechanism have been already published elsewhere [43,44,45]. The starting materials for the ball-milling are denoted F-CNF and F-Gr for carbon nanofibers and KS4 graphite, respectively. After ball-milling, the samples are named F-CNF-Ar, F-CNF-Air, F-Gr-Ar, and F-Gr-Air according to the precursor and the atmosphere of grinding (argon or air).

Works carried out at ICCF on the milling of CFx, when used as a positive electrode in lithium batteries, have led to the patent US20190023574A1. Among the various milling conditions tested, only the ones that result in a significant enhancement of the electrochemical properties are presented here. The grinding conditions chosen for fluorinated graphites and fluorinated carbon nanofibers are as follows. A RETSCH PM100 planetary ball-miller is used. The materials are milled at 350 rpm for 6 h in a 50 mL stainless steel bowl with 10 balls of 10 mm diameter. To avoid excessive heating and thermal decomposition, 36 cycles of 10 min (9 min grinding followed by a one-minute delay) were carried out.

### 2.2. Characterization

The X-ray diffraction (XRD) patterns were recorded using a PANalytical X’PERT X-ray diffractometer with Cu(Kα) radiation. The FTIR experiments were conducted with a Nicolet Summit (Thermoscientific, Waltham, MA, USA) spectrometer in Attenuated Total Reflection (ATR) mode. For each spectrum, 128 scans with 4 cm^−1^ resolution were collected between 4000 and 450 cm^−1^. The ^19^F and ^13^C solid-state NMR experiments were performed using a 300 MHz Bruker Avance spectrometer at room temperature. A cross-polarization (CP)/magic-angle spinning (MAS) NMR probe operating with 2.5 mm and 4 mm rotors was used at 30 kHz and 10 kHz spinning rates for the ^19^F and ^13^C measurements, respectively. A probe (Bruker) with fluorine decoupling on a 4 mm rotor was used. For the MAS spectra, a simple sequence was performed with a single π/2 pulse length of 4.0 and 3.5 μs for ^19^F and ^13^C, respectively. The ^13^C NMR was performed at a frequency of 73.4 MHz and tetramethylsilane (TMS) was used as the reference. The ^19^F NMR was carried out with a frequency of 282.2 MHz and the spectra were externally referenced to CF_3_COOH, then to CFCl_3_ (δ_CF3COOH_ = −78.5 ppm/CFCl_3_). The Transmission Electron Microscopy (TEM) samples were prepared with chloroform and ultrasonication. The Raman spectra were obtained through Invia Kontor (Renishaw, Gloucestershire, UK) equipment and registered at 532 nm and 0.5 mW using 5s as the acquisition time. The spectra resolution was 1.5 cm^−1^.

### 2.3. Electrochemical Tests

An electrochemical study was performed using both cyclic voltammetry and galvanostatic discharge. The positive electrode of the powdered fluorinated compound was composed of the sample (80 wt.%, 2–3 mg), acetylene black (from Mersen, 10 wt.%) to ensure the electronic conductivity, and polyvinylidene difluoride (PVDF powder from Aldrich, with 2–40 μm particle size and average Mw ~534,000 by GPC, 10 wt.%) as a binder. After stirring in propylene carbonate (PC), the mixture was spread uniformly onto a stainless-steel current collector disk of 12 mm in diameter. After PC evaporation, the disk was heated in a vacuum at 40 °C and then 120 °C, for 1 h for each temperature, to remove traces of both water and solvent. The electrolyte was composed of lithium hexafluorophosphate (LiPF_6_) salt dissolved in a mixture of propylene carbonate/ethylene carbonate/dimethyl carbonate (PC:EC:3DMC; 1:1:3 vol%). Three Celgard separators impregnated with the electrolyte were placed between the electrodes, and a lithium foil was used for both counter and reference electrodes. The button cells (CR2032) were assembled in an argon-filled dried glove box. Relaxation was performed for 5 h before any electrochemical process. Then, cyclic voltammetry was carried out at a scan rate of 0.01 mV·s^−1^ in the potential range between 1.5 V and 4.0 V at room temperature. Galvanostatic discharge curves were recorded at 10 mA/g until 1.5 V. Power tests were performed at different discharge rates considering 1C = 751 mA/g. The tests were carried out using an MPG apparatus from Biologic.

## 3. Results

Carbon nanofibres (CNFs) have a closed tubular structure similar to multiwall nanotubes, an average length of 7 ± 2 µm, and an average diameter of 140 ± 30 nm. After fluorination, the length of the nanofibres is reduced. Grinding opens the nanotubes and breaks them into small pieces, resulting in an amorphous structure (Figure 1). Only a few isolated nanotubes remain.

The X-ray diffractograms of fluorinated CNFs before and after grinding are shown in Figure 2. Before grinding, the fluorinated CNF exhibits a typical fluorinated matrix structure with two diffraction peaks at 12.3° and 41.1° in 2θ corresponding to the (001) and (100) diffraction planes, respectively. The (001) diffraction plane corresponds to a d_001_ fluorocarbon spacing of 0.72 nm for the F-CNF. This value is intermediate between 0.9 nm (C_2_F)_n_ and 0.6 nm (CF)_n_ indicating a mixture of these two phases. After grinding, the diffraction peaks on the (001) and (100) planes are less intense for the F-CNF-air and F-CNF-Ar compounds, evidencing a decrease in crystallinity.

Raman spectroscopy was used for the characterization of materials, but photoluminescence quickly took precedence over Raman diffusion. Only for the product F-CNF-air can we distinguish the two characteristic bands of carbon, namely, the band D at about 1350 cm^−1^, and the band G at about 1600 cm^−1^ (see supporting information). It undergoes a lowering of fluorination with air grinding. The covalency of the C-F bonds and the functional groups present in the materials can be probed by infrared spectroscopy. The spectra of the CNFs before and after grinding are shown in Figure 3. All three compounds show a very intense vibrational band located between 1100 and 1200 cm^−1^ corresponding to the elongation of the C-F bonds. The spectrum of the raw F-CNF is characterized by a fine peak centered at 1201 cm^−1^ corresponding to covalent C-F bonds with a slight shoulder at 1108 cm^−1^ attributed to C---F bonds with a weakened covalence. The CF_2_ detected at 1342 cm^−1^ and 962 cm^−1^ are related to the (C_2_F)n phase.

After grinding, the intense peak of covalent C-F bonds shifted slightly (1192 cm^−1^ for the F-CNF-air and 1197 cm^−1^ for the F-CNF-Ar) and broadened towards lower wavenumbers with a peak centered at 1095 cm^−1^ for the F-CNF-air and 1090 cm^−1^ for the F-CNF-Ar. The CF_2_ and CF_3_ groups are still present at 1342 cm^−1^ and 962 cm^−1^, respectively. However, the C=C bonds at 1560 cm^−1^ are more intense after milling, which means that there are more non-fluorinated sp^2^ hybridized carbons after milling. In addition, the overall shape of the spectrum shows a different baseline for the ground materials, slightly approximating the spectrum of a less fluorinated carbon. The F-CNF-Air shows two additional bands: a vibration band at 1750 cm^−1^ corresponding to C=O bonds, and a band at 1872 cm^−1^ for CO-F bonds, carbon–oxygen bonds with an oxygenated environment. These bonds are due to the presence of oxygen during the grinding of F-CNF in air.

The solid-state ^13^C and ^19^F→^13^C cross-polarization MAS NMR spectra are shown in Figure 4. The F/C fluorination rates determined by the fit of the ^13^C spectra are 0.71 for the F-CNF, 0.57 for the F-CNF-air, and 0.71 for the F-CNF-argon. All three compounds have ^13^C NMR spectra with three contributions. The C-F bonds are at 84 ppm for the F-CNF and 83 ppm for the F-CNF-air and F-CNF-Ar. The sp² hybridized carbons interacting with neighboring C-F bonds are located at 135 ppm. Non-fluorinated sp^3^ hybridized carbons give rise to a resonance at 42 ppm. These diamond-like carbons are typical of the (C_2_F)_n_ structural type. The ^19^F→^13^C CP-MAS reveals the presence of CF_2_ groups, which are masked on the ^13^C NMR spectra. CF_2_ groups appear to be more present in the two milled compounds, which contributes to the higher fluorination rate compared with the F-CNF-air.

The ^19^F spectra of the fluorinated CNFs before and after milling are shown in Figure 4c. Whatever the sample, the fine peak centered at −187 ppm is assigned to the covalent C-F bonds. This peak is slightly asymmetric, indicating the presence of a (C_2_F)n phase (C-F weakened by the presence of non-fluorinated C sp^3^). The CF_2_ and CF_3_ groups are present at −116 ppm and −78 ppm, respectively [42,43,44,45]. The ^19^F NMR did not show any major differences between the unground and ground samples.

The ^19^F→^13^C cross-polarization spectra of the CNF compounds before and after grinding (Figure 4b) confirm what has already been stated with the ^13^C and ^19^F NMR. The resonance peak corresponds to covalent C-F bonds at 82.9 ppm for the F-CNF and 83.5 ppm for the milled F-CNF-air and F-CNF-Ar. The CP-MAS NMR highlights the peaks of carbons bonded to fluorine atoms; this is why a slightly higher intensity can be seen for CF_2_ at 110 ppm. Such an observation does not indicate an increase in the number of CF_2_ groups after grinding, as this would be impossible without the addition of fluorine. The graphitic sp^2^ carbons, located at 134.0 ppm, are more present after grinding, particularly after grinding under argon. The sp^2^ carbons provide better electronic conductivity within the material, resulting in a better-operating potential for the cell [37]. These NMR spectra also demonstrate the change in the covalency of the C-F bonds by the broadening of the C-F peak for the ground compounds. Before milling, the width at half-height of this peak is 6.9 ppm for the F-CNF, and this value increases to 9.7ppm for the compounds milled in air and argon.

The electrochemical properties of fluorinated CNFs before and after grinding were studied. As their C-F bonding and structure are similar to that of graphite, with a 2D structure after grinding, the results can be compared with those obtained with fluorinated KS4 (see the physicochemical data of fluorinated graphite in the Appendix A). Cyclic voltammetry was used to highlight the various C-F bonds in the material. The voltammograms of the fluorinated CNF compounds before and after grinding, recorded at scan rates of 0.01 mV/s up to 1.5 V, are shown in Figure 5a.

The F-CNF has a reduction peak centered at 2.48 V, corresponding to the breaking of C-F bonds. After milling, the reduction peak for the F-CNF-Air is centered at 2.58 V and 2.70 V for the F-CNF-Ar. The potential increases after milling, reflecting either a weakening of the C-F bonds or changes at the electrode/electrolyte interface. These results are in line with what was discussed in the physicochemical characterization of the materials. In addition to the potential, the shape of the reduction peak and its width at half-height provide other information. Before grinding, the reduction peak for the F-CNF has a half-value width of 0.265 V. After grinding in air, this value increases to 0.307 V, meaning that the nature of the bonds is less homogeneous in this compound. In addition, the reduction peak of the F-CNF-Air is asymmetric and can be deconvoluted into two Gaussian components corresponding to either two types of C-F bonds or two electrochemical processes with different activation energies (Figure 5b). In the first hypothesis, the F-CNF-Air would be then composed of 41% covalent C-F bonds with a reduction peak centered at 2.35 V, and 59% weaker C-F bonds centered at 2.59 V. The peak is thinner than after grinding in air, which means that the electrochemical active sites are more homogeneous. After milling under argon, the width at half-height of the reduction peak for the F-CNF-Ar is 0.267 V. The fit of the reduction peak leads to 71% of weakened C-F bonds being centered at a potential of 2.69 V and 29% of stronger bonds at 2.49 V. The hypothesis of the two types of C-F bonds is not corroborated by the physico-chemical characterization; neither IR spectroscopy nor NMR data evidence weakened C-F bonds in these high amounts (59 and 71%). Although close C-F bonding, structures, contents of CF_2_ groups, and morphologies are known for the milled CNF, the discharge potentials depend on the atmosphere of milling. It is known that the electrochemical behavior of covalent fluorinated carbons is strongly related to their insulating character at the beginning of the process and the accumulation of LiF on the sheet edges for a higher depth of discharge (DoD). It is important to deeply investigate the galvanostatic curves for the very low DoD (Figure 6a,b). Due to the sub-fluorination, a small potential drop (0.2 V) is recorded for the F-CNF. The over-potential, typical of CFx, is suppressed after milling. The electrochemical process is even initiated at a higher potential for the F-CNF-Air (3.2 V). This enhancement is explained by the presence of conductive sp^2^ carbons of two origins: (i) those formed by the milling, by IR data, and (ii) those already present in the fiber core because of the sub-fluorination and made accessible for the electrolyte by the milling. The defluorination of CFx was evidenced during the tests to establish their tribological properties, in particular for the F-CNF [45]. Back to the physicochemical characterization, the only difference appears on IR spectra, with the presence of oxygenated groups (C=O) when the milling is carried out in the air (Figure 3). It is important to note that the suppression of the over-potential is not achieved with micrometric F-Gr milled in argon (Appendix A).

The galvanostatic discharges at a current density of 10 mA/g up to 1.5 V of the compounds before and after grinding are shown in Figure 6. The three compounds show a galvanostatic discharge with a classic CFx profile with a single plateau, which demonstrates the homogeneity of the samples in terms of C-F bonds and electroreduction kinetics within the material. Otherwise, several plateaus with different potential values would be present. Before grinding, the F-CNF delivers a capacity of 759 ± 38 mAh/g (C_theoretical_ = 747 mAh/g). After grinding, the capacities are 741 ± 38 mAh/g and 743 ± 38 mAh/g for the F-CNF-Air (C_theoretical_ = 670 mAh/g) and the F-CNF-Ar (C_theoretical_ = 746 mAh/g), respectively. The delivered capacities are then those expected theoretically, which means that all the C-F bonds participate in the discharge reaction: CFx + xLi^+^ -> xLiF + C. The average reduction potentials of the materials are 2.65 V for CNF-raw, 2.72 V for CNF-air, and 2.70 V for CNF-Argon. The rather high potential of the F-CNF despite the covalent C-F bonds is caused by the sub-fluorination of the CNFs (F/C = 0.71), which retain a proportion of non-fluorinated sp^2^ Cs that provide better electronic conduction than fully fluorinated CNF (F/C = 1). The evolution of the average reduction potential before and after milling follows the same trend observed by cyclic voltammetry: the potential is higher after milling.

To study the influence of grinding on electrochemical performance, the F-CNF and the F-CNF-Ar were tested at different discharge rates up to 6C. Applied current densities, obtained capacities (Q_exp_), potentials at half-discharge (E_1/2_), and corresponding specific energy densities are summarized in Table 1 for the F-CNF-raw, and in Table 2 for the CNF-Argon. The galvanostatic discharge curves based upon various current densities of the F-CNF and F-CNF-Ar are represented in Figure 6a–d, respectively.

The discharge curves of CNF-raw exhibit a well-defined plateau, regardless of the discharge regime (Figure 6). The capacities delivered are equivalent to a regime of 0.2C with huge yields of 96%. The capacities decrease slightly to 1C and 2C, the Faradic yields being 89 and 79%. There is no significant capacity at a discharge regime of 6C in the range of potentials studied due to excessive polarization. Concerning the potential, a shift of 0.05 V is observed from the C/20 discharge regime to 0.63 V for 2C. This decrease in potential has a direct impact on the mass–energy density of the battery. Indeed, even if the capacity is maintained when the current density increases, the mass–energy density drops by 22 and 38% to 1C and 2C, respectively, because of a lowered potential. Those data are in good accordance with our previous works [45] in strictly similar electrochemical conditions (close F/C ratio, same electrode composition, electrolyte, and current densities); the best energy density was 1780 Wh/Kg and 98% of the Faradic yield. It is important to note that the electrochemical performances can be compared only with the same operating conditions.

For the F-CNF-Ar, the discharge curves maintain the same profile by increasing the discharge regime (Figure 6c). The resulting capacities do not decrease significantly to 2C. (Table 2) Up to 0.2C, the Faradic yields are 100% and remain above 90% up to 2C. The CNF-Argon delivers a significant capacity of 490 mAh/g at a 6C discharge regime, corresponding to a staggering 67% efficiency. The decrease in the average reduction potential is less marked for CNF-Argon, with only a decrease of 0.37 V to 2C unlike a decrease of 0.63 V for CNF-crude. The potential remains higher for the F-CNF-Ar, thanks to the higher proportion of sp^2^ carbons that promote electron conduction [43,44,45]. The capacity delivered to 2C is also better after grinding under argon, despite the lower fluorination rate. The F-CNF-Ar demonstrates interesting performances and surpasses the data in the literature. For comparison, the work of Reddy et al. on the grinding of fluorinated graphite led to improvements that remain lower than those obtained in our work in terms of potential, Faradic yield, and energy density [35]. At a discharge rate of 6C, our CNF-Argon material delivers 67% of its expected capacity, and an energy density of 1083 Wh/kg versus only 40% delivered capacity for Reddy et al. fluorinated graphite, and 800 Wh/kg energy density at 6C. In addition, their material shows a significant decrease in potential to 2.04 V from 1C. In our case, the potential is 2.43 V to 1C, and its decrease is less, since, at 6C, the value of the potential is maintained at 2.21 V. Our material also demonstrates good electrochemical performance under less favorable experimental conditions than some works. Indeed, working at room temperature could be a brake to good performance, as a higher temperature improves electronic conduction. However, our results remain superior to work performed at higher temperatures where Zhou et al. [34] obtained potentials of 2.11 V and 2.23 V at 35 °C and 55 °C, respectively, at a current density of 3000 mA/g. In our study, the potential was measured at 2.21 V at room temperature for a current density of greater than 4000 mA/g.

The fluorinated carbon nanofibers, when ground, delivered a maximum power density of 9693 W/Kg, which is higher than the fluorinated nanofibers 8057 W/kg in LiBF_4_ in PC:DME (1:1 vol.) electrolyte [43]. The overall performance positions them favorably compared to other results in the literature. The nanometric size of the sub-fluorinated F-CNF is necessary for such enhancements. As a matter of fact, for the case of F-Gr-Ar, power and energy densities reached only 1521 W/Kg and 937 Wh/kg, respectively (1324 W/Kg and 1950 Wh/kg for F-Gr, see Appendix A).

Carbon nanofibers had initially been chosen for their sub-fluorination capacity (CF_0.71_), namely, a non-fluorinated core that facilitates the flow of electrons during the electrochemical process, and therefore the power density. These core sp^2^ carbons do not disappear during grinding and are supplemented by a second level of sub-fluorination with sp^2^ C created by grinding (Figure 7). In addition, the structure, which initially consisted of multi-walled tubes, is «opened» by grinding, with two benefits: (i) easier diffusion of Li^+^ and F^−^ ions due to the nanometric size of the CFx and (ii) the formation of LiF distributed on the many plane edges, facilitating electroreduction. Both the Faradic yield and power benefit from this particular configuration.

## 4. Conclusions

The milling of micrometric sub-fluorinated graphite (KS4) and nanometric sub-fluorinated carbon nanofibers did not lead to the same changes, depending on their size. The grinding of graphite rather altered its electrochemical performance at higher current densities. In the case of CNF, similar milling was beneficial under argon because it improved performance in terms of the potential (2.70 V at 0.01C for the F-CNF-Ar versus 2.65 V at 0.01C for the raw F-CNF), Faradic yield (67% at 6C whereas the galvanostatic curve cannot be recorded for the raw F-CNF), energy (1083 Wh/Kg at 6C for the F-CNF-Ar), and power densities (9693 W/Kg for the F-CNF-Ar versus 3192 W/Kg for the raw F-CNF). Using these materials, the results were superior to most of those already reported in the literature. The enhancements can be explained neither by a change in the C-F bonding nor by drastic chemical changes. The electrode/electrolyte interface is probably more favorable for fast ion exchange (Li^+^ and F^−^). This is well evidenced at a very low depth of discharge with the suppression of the over-potential, which is typical of conventional CFx. Other sub-fluorinated nanocarbons would be considered to extend the proposed concept of dual sub-fluorination, e.g., graphitized carbon blacks or nanodiscs. Although the energy and power densities do not exceed the best current performances, e.g., 92.5 kW/kg in [7], 72.9 kW/Kg in [10], and 24.364 kW/kg in [23], the comparison under strictly similar electrochemical operating conditions between the initial fluorinated material and the milled one highlights benefits. This method could be used for a variety of materials [8,15]. Some improvements in the electrolyte composition (e.g., [23,25]) could also be used for milled fluorinated nanofibers.

## Figures and Tables

**Figure 1 nanomaterials-14-00404-f001:**
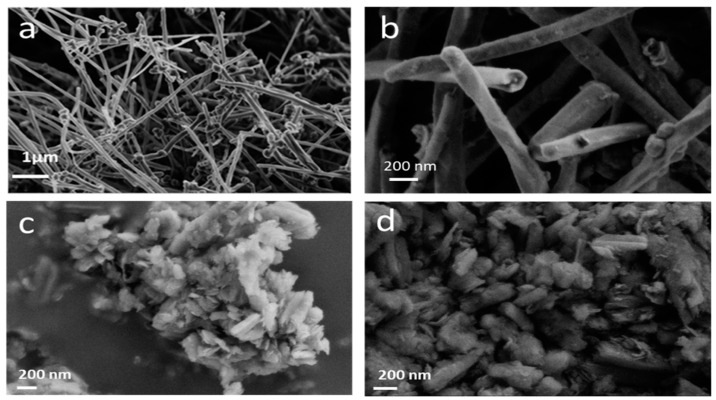
SEM images of CNFs before fluorination (**a**), after fluorination (**b**), and fluorinated after grinding in air (**c**) and in argon (**d**).

**Figure 2 nanomaterials-14-00404-f002:**
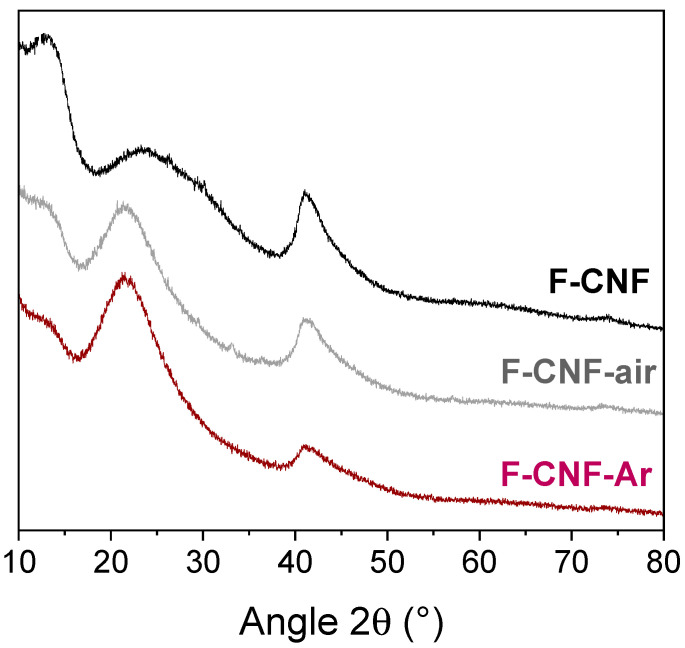
X-ray diffractograms of fluorinated CNF compounds before and after grinding in air and argon atmosphere. The broad line centered at 23° is related to the substrate for XRD measurements.

**Figure 3 nanomaterials-14-00404-f003:**
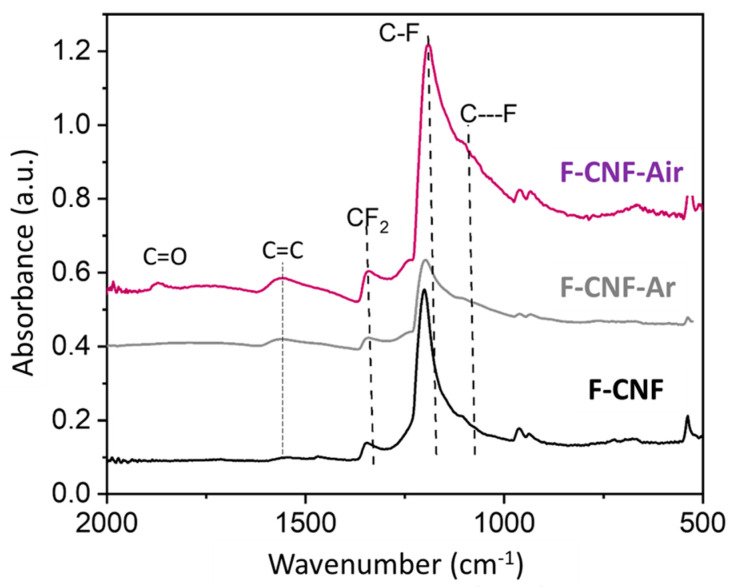
Infrared spectra of fluorinated CNF compounds before and after milling.

**Figure 4 nanomaterials-14-00404-f004:**
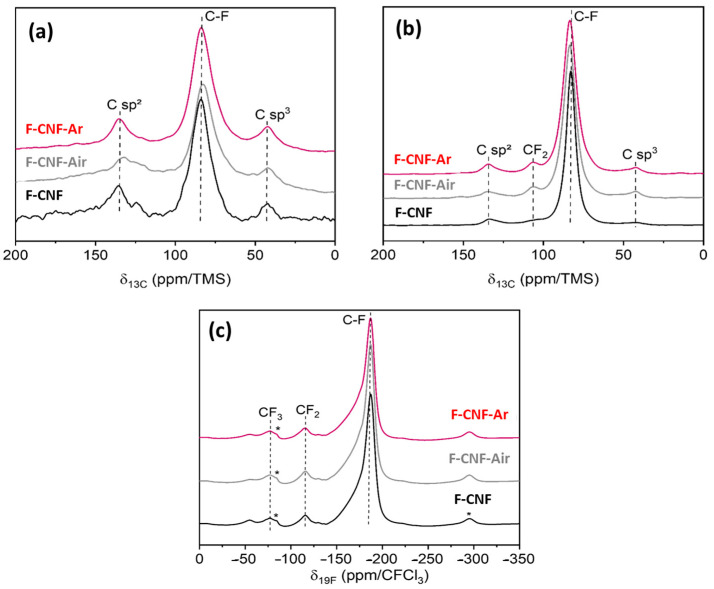
^13^C NMR (**a**), ^19^F→^13^C CP (**b**), and ^19^F (**c**) MAS spectra of fluorinated CNFs before and after grinding. * are spinning sidebands

**Figure 5 nanomaterials-14-00404-f005:**
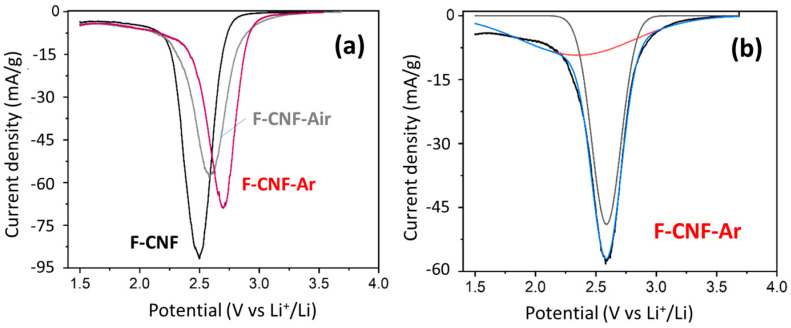
Voltammograms of fluorinated CNF before and after grinding recorded with 0.01 mV/s up to 1.5 V (**a**); an example of fit of the F-CNF-Air voltammogram into two Gaussian components (**b**).

**Figure 6 nanomaterials-14-00404-f006:**
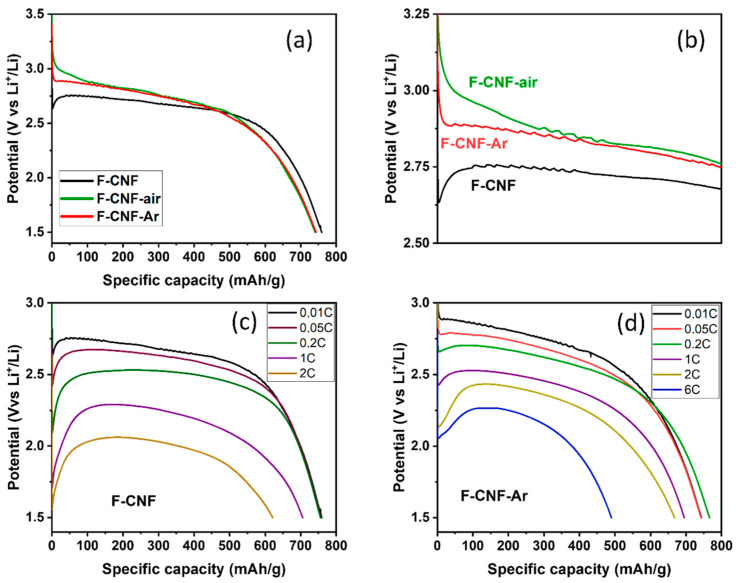
Galvanostatic discharge curves at 10 mA/g up to 1.5 V of fluorinated CNF before and after grinding with LiPF_6_ 1M in EC/PC/3DMC with a current density of 10 mA/g (**a**,**b**) (magnification of the initial part, at different discharge rates (**c**,**d**) for F-CNF and F-CNF-Ar, respectively.

**Figure 7 nanomaterials-14-00404-f007:**
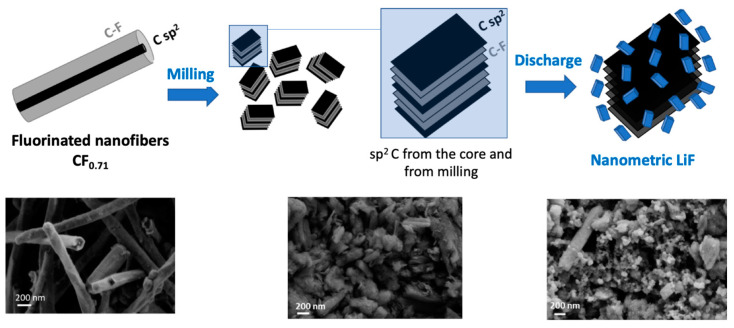
Schematic view of the dual sub-fluorination caused by initial fluorination (the core of the fibers) and milling (creation of C sp^2^ on the surface of the formed nanoparticles).

**Table 1 nanomaterials-14-00404-t001:** Electrochemical data of F-CNF obtained at different reducing current densities.

Current Rate	Current Density (mA/g)	C_exp_ ± 10% (mAh/g)	E_1/2_ (V)	Specific Energy (Wh/kg)	Power Density (W/kg)	Faradic Yield (%) ± 10%
0.01C	10	759	2.65	2011	26	96
0.05C	40	757	2.60	1968	104	96
0.2C	158	756	2.51	1898	396	96
1C	790	706	2.23	1574	1762	89
2C	1580	621	2.02	1254	3192	79

**Table 2 nanomaterials-14-00404-t002:** F-CNF-Ar electrochemical data obtained at different reducing current densities.

Current Rate	Current Density (mA/g)	C_exp_ ± 10% (mAh/g)	E_1/2_ (V)	Specific Energy (Wh/kg)	Power Density (W/kg)	Faradic Yield (%) ± 10%
0.01C	10	743	2.70	2006	27	102
0.05C	36	742	2.63	1951	95	101
0.2C	146	766	2.57	1969	375	105
1C	731	695	2.43	1689	1776	95
2C	1462	668	2.33	1556	3406	91
6C	4386	490	2.21	1083	9693	67

## Data Availability

Data are contained within the article.

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
