# Peer review of "High Energy Density of Ball-Milled Fluorinated Carbon Nanofibers as Cathode in Primary Lithium Batteries"

_nanomaterials, 2024, doi:10.3390/nano14050404_

Round 1

Reviewer 1 Report

Comments and Suggestions for Authors

In this manuscript, the authors investigated the sub-fluorinated nanofibers (F-CNF) and the alterations induced by grinding, providing comprehensive material characterization tests and an analysis of their electrochemical performance. The structure composed of multi-walled tubes was "opened" during grinding, increasing the specific surface area of carbon nanotubes. This facilitated easier diffusion of Li+ and fluoride ions F- and led to some improvement in the electrode performance, showcasing its potential in electrochemical applications.

1The figures could be enhanced for better aesthetics, such as using bolder lines. Additionally, for comparative figures, choosing more distinguishable colors would increase differentiation between different conditions.

2There are some language expression and formatting errors in the manuscript that require further refinement.

3In Figure 2 (XRD), there are several minor peaks; it would be beneficial to apply smoothing techniques for better clarity.

4The authors should elaborate in the electrochemical section on the reasons behind high potential and high energy density.

5The differentiation between table titles and text font in the tables is not clear; using bold fonts could improve clarity.

6The conclusion should outline the performance data post-modification.

Comments on the Quality of English Language

Minor editing of English language required

Author Response

We appreciate the feedback from the editor and reviewers on our manuscript. We have revised the manuscript in line with the comments. Please see the details below.

We thank the reviewer for the valuable comments and suggestions, which will enhance the quality of our paper.

Reviewer 2 Report

Comments and Suggestions for Authors

This manuscript studied the ball milling on the sub-fluorination of carbon nanofibers and their capacity performance. The results are interesting and reliable. But more tests are suggested to be added. Suggestions and questions are given below.

1.     “Sub-fluorinated nanofibers” should be revised as “Sub-fluorinated carbon nanofibers”.

2.     “power density” could be added as a keyword.

3.     “F- ions” in line 44 is suggested to be revised as “F-”.

4.     What are the advantages of lithium batteries compared to other energy storage devices? A short discussion is suggested to be added in the introduction along with some typical references to support the viewpoints, such as Rare Metals 2022, 41 (10), 3432-3445; Journal of Energy Storage 2023, 72, 108509.

5.     How about the ball milling time on the sub-fluorination and capacity performance?

6.     TEM images are suggested to be added to reveal the tubular structure of CNFs.

7.     It is hard to tell the 2D structure from Figure 1 as stated in line 129 “Grinding opens up the nanotubes and breaks them up into small pieces, resulting in a 2D structure”. It would be stricter to describe as amorphous structure other than 2D structure.

8.     More electrochemical tests are required, such as cycle stability and EIS. Please refer to Journal of Alloys and Compounds 2022, 903, 163824.

Comments on the Quality of English Language

Moderate editing of English language is required.

Author Response

(The authors gave the same response as above.)

Reviewer 3 Report

Comments and Suggestions for Authors

The authors synthesised a sub-fluorinated carbon nanofibre (CNF) by ball milling and used it as an anode of LIBs in a half cell. 

They used stainless steel as current collector and its effect on the (dis)charging behaviour should be explained and if possible compared with Cu current collectors.

The material characterisation is good, but the electrochemical characterisation should be extended. 

The results should also be compared and discussed with the reported CNF papers. 

The references are not up to date, no 2023 paper is discussed.

The authors should explain why they used PC for coating materials on current collectors, why not NMP? 

Does PVDF can dissolve in PC?

The writing is not clear and professional, should be checked again.

Comments on the Quality of English Language

by reading abstract, everyone can see the level of English is not perfect. 

Author Response

(The authors gave the same response as above.)

Round 2

Reviewer 3 Report

Comments and Suggestions for Authors

The authors responded to some of the reviewers' comments, but some are not satisfactory, such as discussion of papers from 2023 and even 2024, what is the size of PVDF if want to be used as a binder, should be really small in the case of dispersion. The comparison of the results of this paper with the related publications should be added as a table. The cycle life of cells should be provided.

Author Response

We thank the reviewer for the valuable comments and suggestions, which will enhance the quality of our paper.

About 30 very recent references (2023) have been added and briefly discussed in the introduction and conclusion parts. We apologize because those papers are missing in the previous manuscript.

“Because fluorinated carbons (CFx) exhibit the highest theoretical energy density among current cathode materials for lithium primary battery, extensive efforts are devoted for enhancement of either their energy density, their power density or both (see recent review papers [1-4]. The very recent strategies involve i) considering new carbonaceous precursors for fluorination or innovative fluorination routes [5-14], ii) chemical changes of the CFx either on surface or in the bulk [15-22] and iii) change of the electrolyte composition [23-29].”

“Although the energy and power densities do not exceed the best current performances, e.g. 92.5 kW/kg in [47], 72.9 kW/Kg in [10], 24.364 kW/kg in [23]), the comparison under strictly similar electrochemical operating conditions between the initial fluorinated material and the milled one clearly highlights the benefit. The method could be used for a variety of materials [15,22]. Some improvements about the electrolyte composition (e.g. [23,25]) could be also used for milled fluorinated nanofibers.”

  1. Zhang, S.X.; Kong, L.C.; Li, Y.; Peng, C.; Feng, W. Fundamentals of Li/CFx Battery Design and Application. ENERGY Environ. Sci. 2023, 16, 1907–1942, doi:10.1039/d2ee04179k.
  2. Gao, M.T.; Cai, D.M.; Luo, S.F.; Yang, Y.H.; Xie, Y.; Zhu, L.C.; Yuan, Z.Z. Research Progress in Fluorinated Carbon Sources and the Discharge Mechanism for Li/CFx Primary Batteries. Mater. Chem. A 2023, 11, 16519–16538, doi:10.1039/d3ta02425c.
  3. Liu, W.; Ma, S.; Li, Y.; Wan, B.X.; Wu, C.; Ma, S.D.; Guo, R.; Pei, H.J.; Xie, J.Y. Electrochemical Impedance Spectroscopy Analysis for Lithium Carbon Fluorides Primary Battery. ENERGY STORAGE 2023, 68, doi:10.1016/j.est.2023.107699.
  4. Liu, W.C.; Deng, N.P.; Wang, G.; Yu, R.R.; Wang, X.X.; Cheng, B.W.; Ju, J.G.; Kang, W.M. Fluoridation Routes, Function Mechanism and Application of Fluorinated/Fluorine-Doped Nanocarbon-Based Materials for Various Batteries: A Review. ENERGY Chem. 2023, 85, 363–393, doi:10.1016/j.jechem.2023.06.020.
  5. Hu, Y.H.; Kong, L.C.; Li, W.Y.; Sun, L.D.; Peng, C.; Qin, M.M.; Zhao, Z.Y.; Li, Y.; Feng, W. Fluorinated Microporous Carbon Spheres for Li/CFX Batteries with High Volumetric Energy Density. Commun. 2023, 40, doi:10.1016/j.coco.2023.101607.
  6. Zhang, F.; Lan, Y.Y.; Li, R.J.; Wang, J.L.; Wu, S.X.; Cai, L.J.; Zhao, Y.; Wang, W.L. Boosting the Rate Performance of Primary Li/CFx Batteries through Interlayer Conductive Network Engineering. Mater. Chem. A 2023, 11, 20187–20192, doi:10.1039/d3ta04102f.
  7. Peng, C.; Zhang, S.X.; Kong, L.C.; Xu, H.; Li, Y.; Feng, W. Fluorinated Carbon Nanohorns as Cathode Materials for Ultra-High Power Li/CFx Batteries. SMALL METHODS 2023, doi:10.1002/smtd.202301090.
  8. Cheon, S.; Ha, N.; Lim, C.; Myeong, S.; Lee, I.W.; Lee, Y.S. Fabrication and Electrochemical Characterization of Carbon Fluoride-Based Lithium-Ion Primary Batteries with Improved Rate Performance Using Oxygen Plasma. Chem. Eng. 2023, 34, 534–540, doi:10.14478/ace.2023.1074.
  9. Ha, N.; Lim, C.; Ha, S.; Myeong, S.; Lee, Y.S. Electrochemical Characteristics of CFX Based Lithium Primary Batteries Produced by Carbon Fiber Reinforced Plastic-Derived Waste Carbon Fibers. Chem. Eng. 2023, 34, 515–521, doi:10.14478/ace.2023.1061.
  10. Chen, N.E.; Zhang, G.J.; Chen, H.X.; Yue, H.J. Conductive Carbon-Wrapped Fluorinated Hard Carbon Composite as High-Performance Cathode for Primary Lithium Batteries. COATINGS 2023, 13, doi:10.3390/coatings13050812.
  11. Chen, G.B.; Cao, F.; Li, Z.X.; Fu, J.A.; Wu, B.S.; Liu, Y.F.; Jian, X. Helical Fluorinated Carbon Nanotubes/Iron(III) Fluoride Hybrid with Multilevel Transportation Channels and Rich Active Sites for Lithium/Fluorinated Carbon Primary Battery. Rev. 2023, 12, doi:10.1515/ntrev-2023-0108.
  12. Wang, C.; Teng, J.K.; Chen, X.T.; Wei, J.H.; Shi, B.; Yuan, Z.F.; Li, X.L.; Kang, S.S.; Tang, K.K. Preparation of High-Power Lithium Fluoride Carbon Battery via Microstructural Modulation of Ketjen Black. ENERGY Technol. 2023, doi:10.1002/ente.202300635.
  13. Zhang, Y.Q.; Jiang, J.M.; Zhang, L.; Tang, C.; Tong, Z.K.; Wang, X.M.; Chen, Z.Y.; Li, M.R.; Zhuang, Q.C. BF3-Based Electrolyte Additives Promote Electrochemical Reactions to Boost the Energy Density of Li/CFx Primary Batteries. Acta 2023, 470, doi:10.1016/j.electacta.2023.143311.
  14. Chen, L.; Li, Y.Y.; Liu, C.; Guo, F.F.; Wu, X.Z.; Zhou, P.F.; Fang, Z.W.; Zhou, J. Fluorinated Saccharide-Derived Hard Carbon as a Cathode Material of Lithium Primary Batteries: Effect of the Polymerization Degree of the Starting Saccharide. RSC Adv. 2023, 13, 14797–14807, doi:10.1039/d3ra01695a.
  15. Luo, Z.; Ma, J.; Wang, X.; Chen, D.W.; Wu, D.Z.; Pan, J.; Pan, Y.; Ouyang, X.P. Surface Engineering of Fluorinated Graphene Nanosheets Enables Ultrafast Lithium/Sodium/Potassium Primary Batteries. Mater. 2023, 35, doi:10.1002/adma.202303444.
  16. Wang, N.; Luo, Z.Y.; Zhang, Q.F.; Pan, J.A.; Yuan, T.; Yang, Y.; Xie, S.H. Succinonitrile Broadening the Temperature Range of Li/CFx Primary Batteries. Cent. SOUTH Univ. 2023, 30, 443–453, doi:10.1007/s11771-023-5251-6.
  17. Lim, C.; Ha, S.; Ha, N.; Jeong, S.G.; Lee, Y.S. Plasma Treatment of CFX: The Effect of Surface Chemical Modification Coupled with Surface Etching. CARBON Lett. 2023, doi:10.1007/s42823-023-00597-x.
  18. Zhou, H.; Chen, G.; Yao, L.; Zhang, S.; Feng, T.; Xu, Z.; Fang, Z.; Wu, M. Plasma-Enhanced Fluorination of Layered Carbon Precursors for High-Performance CFx Cathode Materials. Alloys Compd. 2023, 941, doi:10.1016/j.jallcom.2023.168998.
  19. Li, L.Y.; Wu, R.Z.; Ma, H.C.; Cheng, B.B.; Rao, S.Q.; Lin, S.; Xu, C.B.; Li, L.; Ding, Y.; Mai, L.Q. Toward the High-Performance Lithium Primary Batteries by Chemically Modified Fluorinate Carbon with d-MnO2. SMALL 2023, 19, doi:10.1002/smll.202300762.
  20. Tang, C.; Jiang, J.M.; Wang, X.F.; Liu, G.F.; Cui, Y.H.; Zhuang, Q.C. Modifying CF75 Cathode by Ultrathin Carbon Layer to Boosts Electrons Transmission and Discharge Capacity for Lithium/Fluorinated Graphite Primary Batteries. Mater. Lett. 2023, 336, doi:10.1016/j.matlet.2023.133901.
  21. Ma, S.; Liu, W.; Zhang, D.M.; Yang, C.; Luo, Y.; Lou, X.B.; Guo, R.; Wang, Y.; Xie, J.Y. Controllable Solvent Treatment of Fluorinated Graphite for High Power Density and Low Cathode Swelling Lithium Primary Batteries. Eng. J. 2023, 474, doi:10.1016/j.cej.2023.145819.
  22. Cheon, S.; Ha, N.; Lim, C.; Myeong, S.; Lee, I.W.; Lee, Y.S. Fabrication and Electrochemical Characterization of Carbon Fluoride-Based Lithium-Ion Primary Batteries with Improved Rate Performance Using Oxygen Plasma. Chem. Eng. 2023, 34, 534–540, doi:10.14478/ace.2023.1074.
  23. Li, ; Cheng, Z.; Liu, J.L.; Che, L.K.; Zhou, Y.K.; Xu, E.M.; Tian, X.H.; Yuan, Z.Z. Solvation Structure Tuning Induces LiF/Li3N-Rich CEI and SEI Interfaces for Superior Li/CFx SMALL 2023, doi:10.1002/smll.202303149.
  24. Wang, H.; Jiang, J.; Chen, Y.; Wu, Z.R.; Niu, X.B.; Ouyang, C.Y.; Liu, J.; Wang, L. Lithium-Ion and Solvent Co-Intercalation Enhancing the Energy Density of Fluorinated Graphene Cathode. ENERGY Chem. 2024, 89, 208–215, doi:10.1016/j.jechem.2023.10.019.
  25. Zhang, Y.Q.; Jiang, J.M.; Zhang, L.; Tang, C.; Tong, Z.K.; Wang, X.M.; Chen, Z.Y.; Li, M.R.; Zhuang, Q.C. BF3-Based Electrolyte Additives Promote Electrochemical Reactions to Boost the Energy Density of Li/CFx Primary Batteries. Acta 2023, 470, doi:10.1016/j.electacta.2023.143311.
  26. Li, L.; Zhang, S.; Chen, C.; Xu, C.L.; Wang, R.; Wu, M.Q. Potassium Ion Electrolytes Enable High Rate Performance of Li/CFx Primary Batteries. Electrochem. Soc. 2023, 170, doi:10.1149/1945-7111/accaae.
  27. Huo, H.B.; Radhakrishnan, S.; Shaw, L.L.; Nemeth, K. High-Energy and High-Power Primary Li-CFx Batteries Enabled by the Combined Effects of the Binder and the Electrolyte. BATTERIES-BASEL 2023, 9, doi:10.3390/batteries9050268.
  28. Liang, H.J.; Su, M.Y.; Zhao, X.X.; Gu, Z.Y.; Yang, J.L.; Guo, W.; Liu, Z.M.; Zhang, J.; Wu, X.L. Weakly-Solvating Electrolytes Enable Ultralow-Temperature (-80 °C) and High-Power CFx/Li Primary Batteries. CHINA-CHEMISTRY 2023, 66, 1982–1988, doi:10.1007/s11426-023-1638-0.
  29. Li H. et al., Ultrasound assisted wet media milling synthesis of nanofiber-cage LiFePO4/C, Ultrason. Sonochem., 68, 105177, nov. 2020, doi: 10.1016/j.ultsonch.2020.105177.

  1. Peng, C.; Zhang, S.X.; Kong, L.C.; Xu, H.; Li, Y.; Feng, W. Fluorinated Carbon Nanohorns as Cathode Materials for Ultra-High Power Li/CFx Batteries. SMALL METHODS 2023, doi:10.1002/smtd.202301090.

Line 117-118: Provider of PVDF powder isAldrich; its average Mw ~534,000 by GPC).

The previous date about fluorinated nanofibers are discussed.

Lines 309-313: “Those data are in good accordance with our previous works [45] in strictly similar electrochemical condition (close F/C ratio, same electrode composition, electrolyte and current densities); the best energy density was 1780 Wh/Kg and 98% of Faradic yield. It is important to note that the electrochemical performances can be compared only with the same operating conditions.”

Line 357-359: “The fluorinated carbon nanofibers then ground deliver a maximum power density of 9693 W/Kg higher than fluorinated nanofibers 8057 W/kg in LiBF4 in PC:DME (1:1 vol.) electrolyte [43].”

Round 3

Reviewer 3 Report

Comments and Suggestions for Authors

Particle size of PVDF still is not included. Please add it to the manuscript, then the paper can be accepted!

Author Response

We thank the reviewer for the valuable suggestion.

The particle size of polyvinylidene difluoride binder is 2-40 mm. The text has been changed.
